# Berberine Improves Cognitive Impairment by Simultaneously Impacting Cerebral Blood Flow and β-Amyloid Accumulation in an APP/tau/PS1 Mouse Model of Alzheimer’s Disease

**DOI:** 10.3390/cells10051161

**Published:** 2021-05-11

**Authors:** Chenghui Ye, Yubin Liang, Ying Chen, Yu Xiong, Yingfang She, Xiaochun Zhong, Hongda Chen, Min Huang

**Affiliations:** 1Department of Neurology, The Seventh Affiliated Hospital, Sun Yat-sen University, Shenzhen 518107, China; yechh@mail.sysu.edu.cn (C.Y.); cheny769@mail.sysu.edu.cn (Y.C.); xiongy69@mail2.sysu.edu.cn (Y.X.); sheyf3@mail.sysu.edu.cn (Y.S.); zhongxch5@mail.sysu.edu.cn (X.Z.); 2Department of Neurology, Zhuhai People’s Hospital (Zhuhai Hospital Affiliated with Jinan University), Zhuhai 519000, China; liangyubin@stu2017.jnu.edu.cn; 3Department of Traditional Chinese Medicine, The Seventh Affiliated Hospital, Sun Yat-sen University, Shenzhen 518107, China

**Keywords:** Alzheimer’s disease, BBR, cerebral blood flow, β-amyloid

## Abstract

Alzheimer’s disease (AD) is accompanied by β-amyloid (Aβ), neurofibrillary tangles, and neuron cell death, and is one of the most commonly occurring diseases among the elderly. The pathology of AD is complex, involving Aβ overproduction and accumulation, tau hyperphosphorylation, and neuronal loss. In addition, chronic cerebral hypoperfusion (CCH) is ubiquitous in the AD patients and plans a pivotal role in triggering and exacerbating the pathophysiological progress of AD. The goal of this study was to investigate the neuroprotective properties of berberine (BBR) and the underlying mechanism. During the study, BBR was administrated to treat the triple-transgenic mouse model of Alzheimer’s disease (3×Tg AD). To thoroughly evaluate the effects of the BBR administration, multiple manners were utilized, for instance, 3D arterial spin labeling technique, Morris water maze assay, immunofluorescence staining, TUNEL assay, laser speckle contrast imaging, western blotting, etc. The results showed that BBR ameliorated cognitive deficits in 3×Tg AD mice, reduced the Aβ accumulation, inhibited the apoptosis of neurons, promoted the formation of microvessels in the mouse brain by enhancing brain CD31, VEGF, N-cadherin, Ang-1. The new vessels promoted by BBR were observed to have a complete structure and perfect function, which in turn promoted the recovery of cerebral blood flow (CBF). In general, berberine is effective to 3×Tg AD mice, has a neuroprotective effect, and is a candidate drug for the multi-target prevention and treatment of AD.

## 1. Introduction

Alzheimer’s disease (AD) becomes the leading cause of dementia in the elderly. Characterized by chronic progressive neurodegeneration, AD results in progressive cognitive impairment and, ultimately, death. The AD pathology relates to a variety of factors. The most frequent histopathological changes include the formation of neurofibrillary tangles (NFTs) and senile plaque in the brain, accompanied by neurodegeneration [1,2]. Previous research on AD did not deeply explore vascular factors. To be more specific, AD and vascular dementia (VD) were treated as two completely distinct diseases, degenerative diseases and vascular diseases [3,4]. However, emerging evidence implies the contribution of chronic cerebral hypoperfusion (CCH) towards the pathogenesis of AD. To date, researchers observe the reduction of CBF in temporal, parietal and frontal cortices among AD patients [5,6]. It is suggested that the reduction of CBF could be a decisive factor in the pathogenesis of cognitive dysfunction.

Although the occurrence and development of AD are related to many factors, such as Aβ deposition, tau protein hyperphosphorylation, synaptic pathological changes, and mitochondrial function, cerebrovascular factors play an important role in the pathogenesis of AD [7]. Specifically, cerebral microcirculation plays an important role in maintaining the stability of the brain environment and ion homeostasis of neurons, which is the physiological basis of cerebral microvascular in the pathological changes of AD such as nerve fibrosis and plaque formation [8]. The decrease of CBF suggests that the disturbance of microcirculation occurs before the pathological changes of AD and could be a documented event at early stages [9]. Aβ is the core component of senile plaques. Its neurotoxicity causes the death and degeneration of neurons. Additionally, it delivers negative impacts towards synaptic structure, such as destruction and a decrease in number. The ultimate outcomes are cognitive decline and behavioral abnormalities [10]. On the other side, researchers observe an increase in the synthesis of β-APP, and significant improvements in β-secretase BACE1 activity in the brain with the existence of cerebral hypoperfusion [10]. These changes can break the balance between the production and degradation of Aβ, and result in an abnormal deposition of Aβ. Another well-accepted pathological feature of AD is neurofibrillary tangle, which is formed by abnormal phosphorylation of Tau protein [11,12]. The decrease of CBF causes early cell injury, which is supported by a growing number of studies. Subsequently, protein phosphatase activity decreases, and protein kinase activity increases, both of which lead to the hyperphosphorylation of Tau protein [13]. According to latest discoveries, therapies targeting one single protein or pathway have little therapeutic effects on complex diseases, such as AD [14]. Instead, novel efficient therapeutics are more likely to be discovered via targeting several key pathways simultaneously.

Berberine (BBR), also known as berberine hydrochloride, is a natural isoquinoline alkaloid extracted from Rhizoma Coptidis. It has a variety of pharmacological effects, including anti-viral, antibacterial, anti-inflammatory, anti-cancer, hypoglycemic, and lipid regulation [15,16]. Additionally, BBR has potential therapeutic effects on AD through different pathological mechanisms, such as lowering Aβ levels, inhibiting the phosphorylation of Tau protein, anti-oxidation, inhibiting the activity of AchE and MAO, and regulating lipids, hypoglycemic. The studies of BBR regarding its treatment effects towards AD mainly focus on how BBR intervenes with Aβ. To be more specific, it illuminates the neurotoxicity of Aβ, the inflammatory response induced by Aβ, and the production of Aβ. However, little attention has been devoted to the impacts of BBR on the CBF of AD.

Therefore, in the present study, we aimed to investigate the effects of BBR on AD regarding the changes in CBF, and the following impacts towards the cognitive function. Given this focus, a novel AD mouse model was utilized.

## 2. Materials and Methods

### 2.1. Drugs and Reagents

Both BBR (99% purity) for treatment and dimethyl sulfoxide (DMSO) were purchased from Sigma-Aldrich (St. Louis, MO, USA). Antibodies specific to Aβ1–42, DAPI, GFAP, NeuN, caspase-3, CD31, VEGF, N-cadherin, Ang-1 and GAPDH were purchased from Abcam (Cambridge, UK). In situ cell death detection kits were utilized to perform the TdT-mediated dUTPnick-end labeling (TUNEL). All other reagents used in the experiment were reagent-grade. Table 1 shows the antibody information.

### 2.2. Animals and Treatment

The 3×Tg AD mice were purchased from Jackson Laboratory (Bar Harbor, ME, USA) to complete the experimental study by expressing human mutant genes, including APPswe, PS1M146V, and tauP301L. Aβ was detected in the cells of these six-month-old mice [17]. In the treatment group (BBR + 3×Tg) (*n* = 12; male), drinking water containing 100 mg/kg/day berberine was given from four months old for four months until they developed cognitive impairment and typical pathological features. The dosage of berberine in this experiment is based on the previous research report [18]. The other two groups were treated with conventional drinking water without any other addition. The two groups were 3×Tg AD mice (3×Tg) (*n* = 12; male) and male non-transgenic wild-type (WT) mice (*n* = 12). The three groups were all carried out under the same standard laboratory conditions, with a temperature of 22 ± 2 °C and a light/dark cycle of 12 h, and free drinking water and eating. Each cage contained three or four objects. The purpose of this experiment was to minimize the pain suffered by animals, which is reflected in all aspects of the experiment process, such as reducing the number of animals used and using the substitute methods of in vivo technology when feasible. This study has always followed the Animal Care and Institutional Ethical Guidelines in China.

### 2.3. Magnetic Resonance Imaging

#### 2.3.1. Scanning Parameters

The MRI was from GE Healthcare, Milwaukee, Wisconsin, USA, with an eight-channel wrist coil discovery 750 3.0T scanner (GE Healthcare, Milwaukee, Wisconsin, USA). All mice were scanned with the same parameters. Each mouse was anesthetized with 10% chloral hydrate (0.3 mL/100 g) and placed in a supine position, then scanned. In this study, all imaging parameters of 3D ASL series are the same as those reported in previous studies [19]. In other words, 15 slices are scanned in ascending order with slice thickness = 4 mm without gaps. Other core parameters include: field of view = 120 mm × 120 mm, matrix = 512 (points) × 12 (arms); number of excitations = 5, bandwidth = 62.5 kHz, scan duration was 9 min 14 s, labeling duration = 1650 ms, post-labeling delay = 1025 ms, repetition time = 4132 ms and echo time = 11 ms. In addition, MR imaging of the blood vessels in mice was obtained by time-of-flight MR angiography (TOF MRA) and 3D Fast SPGR. The scanning parameters of TOF MRA are as follows: echo time = 3.9 ms, repetition time = 20 ms, field of view = 80 mm × 60 mm, matrix = 320 × 224, number of excitations = 1, bandwidth = 31.2 kHz, and scan duration was 231 s.

#### 2.3.2. 3D Arterial Spin Labeling Technique

CBF in different brain regions of the mice was measured using the 3D arterial spin labeling (ASL) technique. The processes and the parameters of this technique were kept the same as the previous study [19].

### 2.4. Morris Water Maze Task

The learning and memory abilities of all mice were evaluated by a five-day (d) Morris water maze task at the age of eight months [20]. During the five-day evaluation, all mice in the three groups received the same treatment as before. All the equipment and test procedures in the experiment were consistent with the previous research [21]. Specifically, this experiment had a round white metal pool; its diameter was 160 cm and its height was 50 cm. The pool was filled with water of constant temperature (22 ± 1 °C) to 26 cm deep. Then, the water maze software (Tianyancha, Sichuan, China) divided the pool into four quadrants and was equipped with a translucent acrylic platform. The platform was placed in the center of the northwest quadrant and 1 to 2 cm below the water surface, with a diameter of 12 cm and a height of 24 cm.

#### 2.4.1. Spatial Learning Test

All mice were tested for four consecutive experiments and five days of spatial learning. The starting position of the mice was selected in the four quadrants of the pool edge and changed every day, but the position of the platform remained unchanged. In the first part of each experiment, the mice were gently placed in the water with their noses toward the surface, at different starting points (north, south, east, and west) from the wall. It took up to 60 s for each mouse to find the hidden platform. If the escape platform is not found within 60 s, it takes another 30 s to manually guide it to the platform. The escape traces of all mice can be recorded by a camera mounted on the ceiling directly above the pool. The human visual system (HVS) water maze program was used for subsequent analysis of escape latency (Water Maze 3, Actimetrics, Evanston, IL, USA). The researchers did not know the group allocation of each mouse during the experiment; that is, all the experiments were carried out by blind method.

#### 2.4.2. Probe Trial

Short- and long-term memory integration was evaluated by probe test. The experiment was carried out after the last experiment, at 24 h and 72 h, respectively. At the beginning of the experiment, the platform was removed, and then the mice were placed in the quarter quadrant of the pool, which was the position of the previous platform. In each probe test, each mouse had a swimming time of 60 s. The time spent in the quadrant pre-placed with the platform and the time spent in the platform position were recorded to evaluate the short- and long-term memory of the mice.

### 2.5. Immunofluorescence Staining

The vector paraffin slices of the mouse brain were mounted on a 5 mm-thick slide. The slices were pre-treated at a high temperature of 0.01 mol/L citrate buffer (pH = 6.0) for 5 min, then sealed off in PBS with 5% goat serum for 10 min. These slices were further incubated overnight at 4 °C with an anti-resistance. The slices were then sliced with a secondary resistance (1:500 in PBS) and incubated for 1 h at 37 °C.

Aβ1–42, GFAP, NeuN, caspase-3 and CD31 were detected by the second antibody (anti-mouse and anti-rabbit; Alexa Fluor 488 and 695, (Multi Sciences Biotech, Hangzhou, China)) coupled with Alexa fluor fluorescent dye. For each mouse, three equidistant parts were evaluated to cover the entire hippocome. To analyze and quantify immune response areas, these slices were imaged using a fluorescent microscope (Olympus, Japan) and analyzed using ImagePro Plus 6.0 software (Media Cybernetics, Shanghai, China).

### 2.6. TUNEL Assay

An in situ pod cell death detection kit (Roche) was used to detect cell death in brain tissue. In other words, the assay determines apoptosis by detecting DNA breaks caused by apoptotic signal cascades and DNA strand breaks labeled with terminal deoxynucleotidyl transferase.

### 2.7. Laser Speckle Contrast Imaging

Real-time 2D CBF perfusion information was provided by a laser speckle contrast imaging system. Before the experiment, the mice were sedated and hypnotized. A median longitudinal incision was made at the back of the brain, and the skin was pulled to the sides to expose the skull. Then, the skull was placed 10 cm below the scan head to record CBF perfusion in real time.

### 2.8. Western Blot Assay

Before moving onto assays on the brain tissue, all tissue samples were required to be homogenized with lysis buffer plus 1 mM PMSF and protease inhibitor cocktail. A BCA protein assay kit was used to assess protein concentration. The total protein extract from each sample was equal, which was 20 μg per well. These protein extracts were separated by SDS-PAGE and transferred to polyvinylidene fluoride (PVDF) membranes. Five percent fat-free milk was utilized as the blocking agents. Afterwards, CD31, VEGF, N-cadherin, Ang-1 and GAPDH were probed with the corresponding primary antibodies (1:1000), followed by the incubation with HRP-conjugated anti-rabbit antibody or HRP-anti-mouse antibody. These blots were developed using ECL detection reagent and visualized as Kodak Image Station 4000 mm (Carestream Health Inc., New Haven, CT, USA). Quantitative software (National Institutes of Health, Bethesda, MD, USA) was used to quantify the characteristics of all frequency bands.

### 2.9. Statistical Analysis

The statistical analysis was carried out using Statistical Product and Service Solutions (SPSS) 17.0 software (SPSS Inc., Chicago, IL, USA). Comparison between groups was analyzed by one-way ANOVA test followed by post-hoc test (least significant difference). *p* < 0.05 was set to be statistically significant.

## 3. Results

### 3.1. BBR Accelerates the Recovery of CBF in the Brain of 3×Tg AD Mice

The 3D ASL technique was applied to detected CBF in every mouse across the three groups. The color signals in the image, ranging from green to red, represent an incremental scale of CBF level. Compared with WT group, most areas in 3×Tg AD group demonstrated blue or green, suggesting an evident lower CBF in brain in 3×Tg AD group. The BBR treatment group (BBR + 3×Tg) showed a relatively higher CBF after four months of administration, compared with the 3×Tg AD group of the same age. The dynamic changes in CBF across different group were further analyzed in a quantitative manner. This difference between the BBR treatment group and 3×Tg AD group was statistically significant (Figure 1C,D). Between the WT group and 3×Tg AD group, the latter showed a significantly lower CBF (Figure 1D, *p* < 0.01). In the comparison between the BBR + 3×Tg AD group and 3×Tg AD group, the treatment group demonstrated a significantly higher CBF (Figure 1D, *p* < 0.01). It supports the hypothesis that BBR could accelerate the recovery of CBF in a 3×Tg AD mouse model.

### 3.2. BBR Relieves Cognitive Impairments in 3×Tg AD Mice

The Morris water maze test assessed spatial learning by measuring the time each mouse spent to find the hidden platform (also known as escape latency). The length of escape latency is supposed to be negatively correlated with the level of spatial learning ability. Additionally, in post hoc multiple comparisons, no significant differences were found in the swimming speed of all groups (Figure 2B, *p* > 0.05). Thus, the results from the Morris water maze test could solidly reflect the effect of BBR on memory impairment in 3×Tg AD mice. In a five-day training trial, compared with WT mice, 3×Tg AD mice exhibited a significantly longer escape path and remarkably longer escape latency to find the platform (Figure 2A). The escape latency of the BBR-treated 3×Tg AD mice was significantly shorter than that in 3×Tg AD mice (Figure 2A). According to the probe trial assay results, the platform span of 3×Tg AD mice treated with BBR was shorter than that of 3×Tg AD mice (Figure 2C,D). Moreover, comparison of the swimming tracts across the three groups suggests that tracts from the BBR treated 3×Tg AD mice group match with those from the WT group better than the 3×Tg AD group (Figure 2E).

### 3.3. BBR Reduces the Production of Aβ and Inhibits Apoptosis in the Brains of 3×Tg AD Mice

The effect of BBR on Aβ burden in the brain was also investigated in the 3×Tg AD mice. The results of immunofluorescent staining of β-amyloid in the hippocampus of mice from all three groups were shown in Figure 3A. As expected, there was little aggregation of β-amyloid found in the WT mice, while the β-amyloid aggregation was apparent in the 3×Tg AD and BBR-treated 3×Tg AD groups (Figure 3A). Moreover, between the latter two groups, the density of β-amyloid aggregating in the hippocampus was remarkably higher in the 3×Tg AD group, and the labeling for β-amyloid was also more intense (Figure 3B, *p* < 0.01).

Immunofluorescence was applied to examine the number of GFAP in the hippocampus of each group. Based on the outcomes shown in Figure 3C, a small number of GFAP were observed in the hippocampus of the WT group. Nevertheless, a noticeable amount of GFAP were observed in the hippocampus of the 3×Tg AD group, with no BBR administration. The amount of GFAP was significantly lower in the 3×Tg AD group with BBR administration, compared with the 3×Tg AD group without administration (Figure 3D, *p* < 0.01). Neuronal apoptosis in hippocampus was also taken into consideration. Immunofluorescence was used to evaluate the apoptosis level, via detecting the expression of NeuN, Caspase-3 and TUNEL. Compared with the WT group, a significant increase in the expression of caspase-3 in the hippocampus of the 3×Tg AD group, as well as a meaningful decrease in the expression of NeuN was observed (Figure 3E, *p* < 0.01). In line with the previous findings, BBR administration relieved these changes. In the BBR-treated 3×Tg AD group, compared with the 3×Tg AD group, the expression of caspaes-3 in the hippocampus was significantly lower (Figure 3G, *p* < 0.01), and the expression of NeuN was significantly higher (Figure 3F, *p* < 0.01), both of which were closer to the level observed in WT group. As shown in Figure 3H, the number of TUNEL-positive cells in the brain tissue of 3×Tg AD mice was larger than that of the WT mice (Figure 3H, *p* < 0.01). Merely considering the two 3×Tg AD groups, with and without BBR treatment, the treatment group demonstrated a significantly less TUNEL-positive cells (Figure 3I, *p* < 0.01) in the brain.

### 3.4. BBR Increases the Cerebral CBF Perfusion and Proangiogenic Factors in the Brains of 3×Tg AD Mice

The cerebral CBF perfusion was detected via laser speckle contrast imaging. According to the laser speckle contrast imaging, the CBF perfusion in the WT mice group was obviously higher than that in 3×Tg AD mice group (Figure 4A). The difference was statistically significant. Compared with the 3×Tg AD mice group, CBF in the BBR-treatment 3×Tg AD group was significantly higher (Figure 4B, *p* < 0.01). It was well-indicated that BBR treatment could increase the blood perfusion in brain tissue. It can be inferred that BBR promotes the formation of blood vessels, and the new vessels have blood flow through, and it is functional blood.

CD31 plays an important role in the formation of blood vessels. It can also be used as a manner to quantitatively evaluate the role of angiogenesis factors in angiogenesis. As a result, it has become an indicator of common angiogenesis in clinical practice [22]. We tested the number of CD31 to determine whether there was angiogenesis or not. Figure 4C shows the number of CD31-positive staining. It demonstrated that staining in 3×Tg AD group was significantly lower than that in WT group (Figure 4D, *p* < 0.01). Compared with 3×Tg AD group, the quantity of CD31-positive staining in the hippocampus increased significantly in the BBR-treated group (Figure 4D, *p* < 0.01). Western-blot was used to detect the expression of CD31, VEGF, Ang-1 and N-Cadherin in the hippocampus of mice, which was further used to evaluate angiogenesis in the hippocampus (Figure 4E). Compared with WT group, the expression of CD31, VEGF, Ang-1 and N-cadherin in the hippocampus of 3×Tg AD group was significantly decreased (Figure 4F, *p* < 0.01). Comparing the outcomes between 3×Tg AD group and BBR + 3×Tg AD group, the latter exhibited significantly higher levels in the expressions of CD31, VEGF, Ang-1 and N-cadherin in the hippocampus (Figure 4F, *p* < 0.01).

## 4. Discussion

Although BBR is a well-known neuroprotective agent [15,16], few studies have focused on whether BBR can promote the recovery of CBF in AD mice. In this experiment, we used a 3×Tg AD mouse model and gave a four-month BBR administration to the treatment 3×Tg AD mice group. We used the ASL scanning method, Morris water maze test, immunofluorescence assay, TUNEL, western blotting and laser speckle contrast imaging to conduct a thorough evaluation of how BBR impacts the cognitive function of 3×Tg AD mice, and which pathways were involved. The outcomes of our study suggest that BBR has a beneficial effect in Alzheimer’s disease via different mechanisms. First, BBR ameliorates cognitive deficits in the 3×Tg mouse model of Alzheimer’s disease. Second, BBR not only decelerates the production of Aβ, but also inhibits the apoptosis of neurons. Last but not least, BBR plays a positive role via promoting the formation of microvessels in the mouse brain, which leads to the recovery of CBF in 3×Tg AD mice.

Research from various aspects, including epidemiology, clinical medicine, neuropathology, throw more light on the correlation between the reduction of blood flow in the brain and the pathogenesis of cognitive dysfunction [3,23]. From the perspective of medical imaging, studies also provide solid evidence that AD has insufficient perfusion in the temporal parietal (including hippocampus), hippocampus [24], and posterior cingulate gyrus [25]. The level of blood flow level in the hippocampus is recognized as the basis for the diagnosis of early AD [26]. Proved by one previous study, after 25 weeks, the mice’s bilateral common carotid arteries being ligated, chronic cerebral ischemia could cause βAPP cleavage to form Aβ fragment. Other observed consequences include extracellular amyloid deposition, as well as some other pathological changes similar to AD [27]. More importantly, these known patterns formulate a vicious loop, which remarkably speeds up the pathological process. Aβ, deposited on the basement membrane of capillaries and perivascular drainage arteries, enhances the endothelial contracture function secreted by endothelial cells. It contributes to cerebral vasoconstriction, further aggravating hypoperfusion. This secondary hypoperfusion accelerates the deposition of Aβ [11]. In conclusion, both chronic hypoperfusion and deposition of Aβ represent causal factors and reinforce each other. They simultaneously accelerate the pathological process of AD together.

With a long history of being used as a monomer in traditional Chinese medicine, BBR is recognized as safe for clinical use. Research has proven its beneficial impact to decrease the CBF and relieve cognitive deficits in rats with chronic cerebral hypoperfusion [28,29]. In our study, we explored if BBR had a similar effect on CBF and behavioral deficits in 3×Tg AD mice. The utilized AD mouse model showed a significant decrease of CBF, which is slowly progressive, compared to the WT group. The observed CBF decrease was partly recovered by BBR administration according to the comparison between 3×Tg AD mice and BBR + 3×Tg AD mice (Figure 1). Additionally, analysis of behavior tests showed better cognitive performance in BBR treatment 3×Tg AD mice at eight months (Figure 2). It suggests that BBR has potent influences on improving CBF and cognitive deficits in 3×Tg AD mice.

Next, we explored the effect of BBR administration on Aβ and apoptosis changes among 3×Tg AD mice. It is well-established that Aβ has neurotoxic effects, which contributes to neuronal degeneration, death, apoptosis, synaptic destruction and number reduction [30]. The eventual outcomes include cognitive decline and behavioral abnormalities [10]. In accordance with previous research, our findings suggest significant increases in GFAP and neuronal loss, as well as a significant decrease in NeuN in the brain of 3×Tg AD mice (Figure 3). In the pathological state, Astrocyte AS is activated by Aβ and other toxic substances, which can produce cytoinitis factors, complements, oxygen free radicals, etc. and initiate inflammatory reactions, promote the damage and death of nerve cells, and aggravate the process of AD [31]. Astrocyte pathological changes can cause damage to vascular regulation and small arterial contraction and hypoxia of tissues, which play an important role in the pathogenesis of AD [32]. The experimental study showed that compared with the WT group, the expression of GFAP in the hippocampi of the 3×Tg AD group increased significantly, indicating that the damage to hippocup tissue in the 3×Tg AD group was obvious, the ascension was active compared with the 3×Tg AD group, and the expression of hippocome GFAP in the berberine group was significantly reduced, indicating that the berberine group inhibited the excessive growth of AD mice AS. This may be related to the improvement of local blood supply by berberine, reducing the toxicity of Aβ, and thus repairing damage to nerve vascular units. It is probably due to the neurotoxic effects of Aβ. Our study supports that BBR has a strong neuroprotection on ameliorating neuronal loss in 3×Tg AD mice at 9 months (Figure 3). Meanwhile, BBR reduces Aβ production and relieves cell apoptosis.

The decrease of CBF and Aβ deposition reinforce each other and accelerate the pathological process of AD [11]. Therefore, we posited that the beneficial role of BBR towards neuronal loss and neurodegeneration in 3×Tg AD mice also related to microvascular formation in the brain.

To verify the above hypothesis, we examined the effect of BBR on CD31, Ang-1, VEGF, N-cadherin expression in 3×Tg AD mice at eight months. CD31 plays an important role in angiogenesis and can be the specific marker for the identification of vascular endothelial cells. It could also provide quantitative assessment of vascular correction factors in angiogenesis [24]. CD31 is proved to have the capacity of formulating micro vessels and establishing collateral circulation in the mouse brain. Another key function of CD31 is that it facilitates the repair of nerve cells [33]. According to our results, after the administration of BBR in 3×Tg AD mice, the expression of CD31 was significantly increased. It implies that BBR is able to induce angiogenesis and strengthen the nutritional supports towards the brain. Vascular endothelial growth factor (VEGF) can directly act on neurons in the central nervous system. The interacting mechanism could be taken in multiple forms, for examples, neurotrophic, neuroprotective, anti-apoptotic, or in the process of cell proliferation. Similar with the CD31, the level of VEGF is relatively lower in AD patients compared with the healthy population. Such change delivers inhibitory effects on cerebral angiogenesis, neuronal protection, cerebral microvascular trophic factors, and permeability, which in turn aggravates the damage of the brain with the existence of hypoxia [34]. The impact of VEFG also reflects the pathological progress of AD. A negative correlation between the level of VEGF and the development progress of AD’s pathological process has been observed [35]. VEGF can act directly on many types of nerve cells to play a neurotrophic and neuroprotective role, enhance cell activity and survival, and promote axon regeneration. The experimental study shows that compared with the WT group, the expression of VEGF in the hippocampi of the 3×Tg AD group mice was significantly decreased, indicating that 3×Tg AD group mice have vascular damage and neovascular regeneration disorders, thus leading to nerve vascular unit damage and brain vascular reserves, poor microcirculation blood supply, and thus cognitive dysfunction, which may be related to the nerve and vascular toxicity of Aβ. Compared with the 3×Tg AD group, the VEGF in the Berberine group was significantly increased, which was related to berberine-promoting angiogenesis and the acceleration of the removal of toxic substances such as Aβ by the tissue structure around microvascular vessels. Moreover, angiogenesis is necessary for local tissue metabolism and functional recovery, which can increase cerebral blood flow and improve learning and memory ability. Another factor taken into consideration is Ang-1. Ang-1 is mainly produced by pericytes. It plays a positive role in angiogenesis by binding to the Tie-2 reception on the endothelium [36]. In addition to ensuring the survival of endothelial cells, Ang-1 promotes the maturation of new vessels [37]. It was found in this experiment that the expression of Ang-1 and N-cadherin were meaningfully higher in 3×Tg AD mice at eight months after treatment with BBR, compared to 3×Tg AD mice with no BBR treatment. N-cadherin is another substance facilitating vascular maturation, realized by promoting the encapsulation of pericytes on endothelial cells [38]. This finding suggests that BBR not only promotes angiogenesis but also facilitates the maturation of new vessels. The latest research finds N-cadherin as a key signaling protein which has the potential to open up the anti-apoptotic pathway of nerve cells [39]. According to the immunofluorescence results of CD31, the BBR-treated group was significantly elevated, indicating that BBR plays an important role in the maturation of new vessels.

Our studies provided evidence from different aspects that BBR could promote the increase of cerebral blood perfusion in 3×Tg AD mice. It is indicated that berberine promoted angiogenesis, which improved blood perfusion of brain tissue. The underlying mechanisms of how BBR improves cognitive impairment and alleviates neuronal loss could be numerous, which include decreasing Aβ accumulation, accelerating angiogenesis, and recovering cerebral blood perfusion.

## 5. Conclusions

The BBR can clearly improve the learning and memory abilities of the AD model mice. The BBR’s protection on AD may be connected with promoting CBF recovery and repressing the pathology of AD. Taking all these results into consideration, it could be concluded that BBR could promote angiogenesis, accelerate the elimination of toxic substances such as AB in the tissue structure around microvessels, and inhibit the apoptosis of nerve cells. These results suggest that BBR could have the potential for the therapy of Alzheimer’s disease through promoting CBF recovery.

## Figures and Tables

**Figure 1 cells-10-01161-f001:**
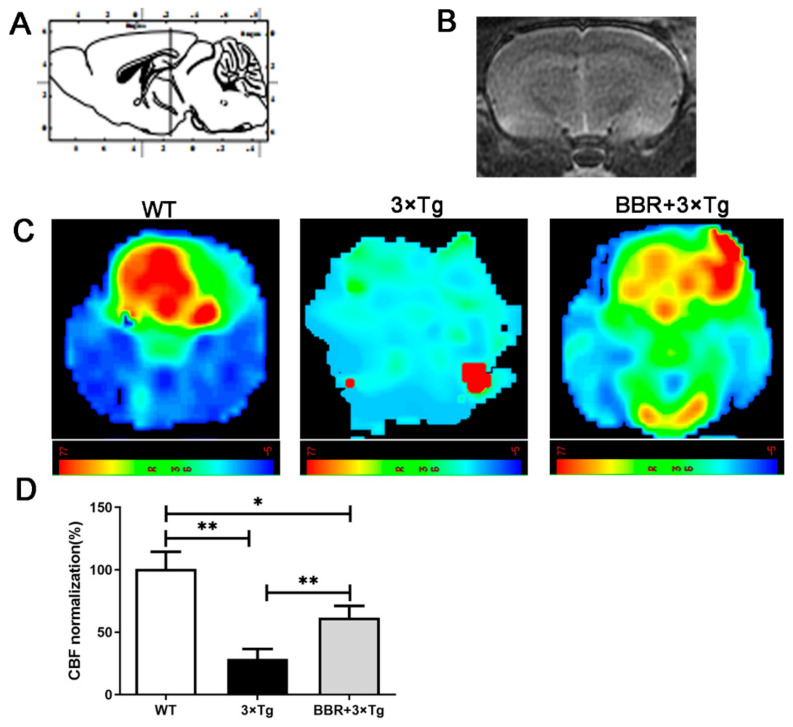
Magnetic resonance imaging (MRI; 3D ASL) analysis showing changes in CBF in the brain of mice. (**A**) A cross-sectional image corresponding to the brain mapping of mice. (**B**) The gray-scale images are T2 imaging. (**C**) MRI images show the changes in CBF in the brain of mice, and color images are those with 3D ASL. The red label indicates a relatively higher CBF level, while the green indicates a relatively lower CBF level. (**D**) Quantitative analysis results of CBF in mice brains. WT, wild-type mice, 3×Tg, 3×Tg AD mice, BBR + 3×Tg; 3×Tg AD mice were given BBR at 100 mg/kg/day. One-way ANOVA and Dunnett’s post-hoc test were used to analyze all the data, expressed as mean ± SD; *n* = 12 animals/group. * *p* < 0.05, ** *p* < 0.01.

**Figure 2 cells-10-01161-f002:**
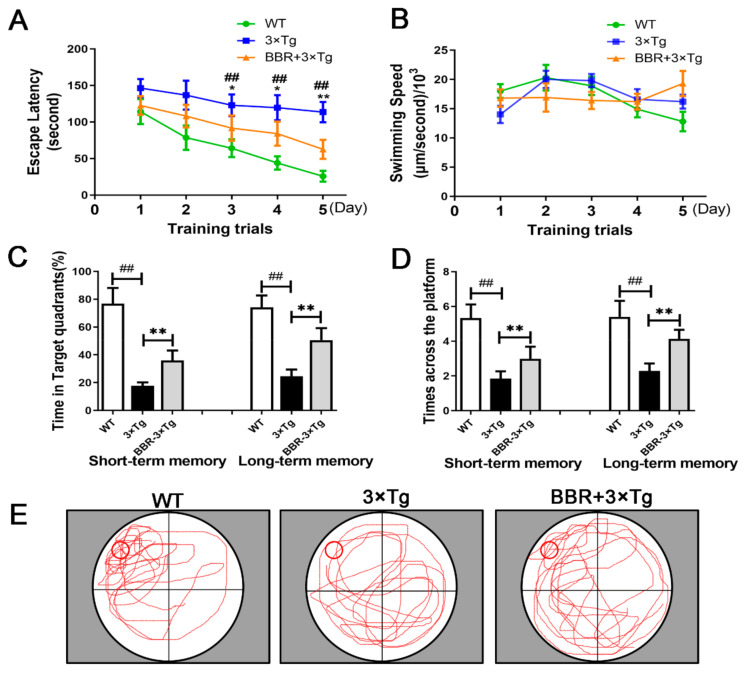
A Morris water maze task was used to evaluate the effect of berberine on the spatial learning and memory of 3×Tg AD mice. (**A**) In five-day training trial, the escape latency of each mouse was measured. (**B**) There were no significant differences in swimming speed across the three groups. (**C**) In the probe trail, the frequency at which the mouse passed through the submerged platform placement area during the training trial was recorded. (**D**) Time spent searching for the pre-placed platform in the target quadrant, in both the short- and long-term memory tests. (**E**) The swimming tracks of the mice from the three groups made in the water tank on the last day of the test. WT, wild-type mice, 3×Tg, 3×Tg AD mice; BBR + 3×Tg; 3×Tg AD mice were given BBR at 100 mg/kg/day. One-way ANOVA and Dunnett’s post-hoc test were used to analyze all the data, which are presented as mean ± SD; *n* = 12 animals/group. * *p* < 0.05, ^##^ *p* < 0.01, 3×Tg AD group vs. WT group. * *p* < 0.05, ** *p* < 0.01, BBR treated 3×Tg vs. 3×Tg group.

**Figure 3 cells-10-01161-f003:**
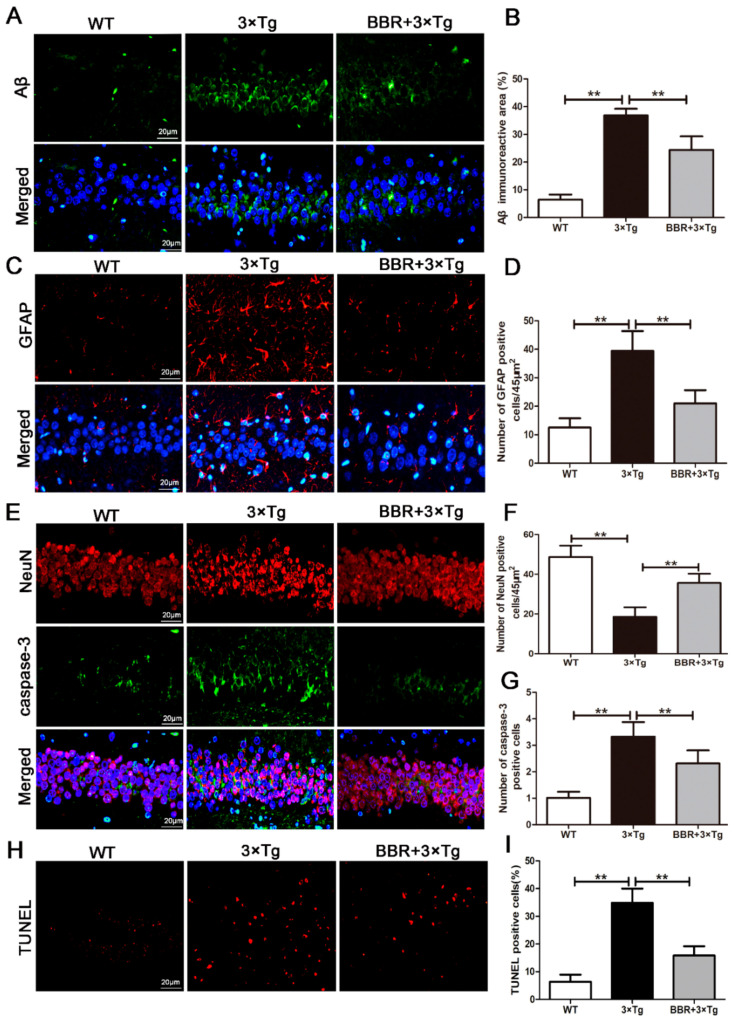
BBR reduced the production of Aβ and inhibited apoptosis in the brains of 3×Tg AD mice. (**A**) Immunofluorescence of Aβ (green) in the hippocampi of mice from the three groups. Scale bar: 20 μm. (**B**) Quantification of Aβ immunoreactivity in the hippocampi of mice. (**C**) Immunofluorescence of GFAP (red) in the hippocampi of mice from the three groups. Scale bar: 20 μm. (**D**) Quantification of GFAP immunoreactivity in the hippocampi of mice. (**E**) Immunofluorescence of NeuN (red) and Caspase-3 (green) in the hippocampi of mice from the three groups. Scale bar: 20 μm. (**F**) Quantification of NeuN and Caspase-3 immunoreactivity in the hippocampi of mice. (**G**) Quantification of Caspase-3 immunoreactivity in the hippocampi of mice. (**I**) The number of TUNEL-positive neurons in each group was averaged. (**H**) The number of TUNEL-positive neurons (red). Scale bar: 20 μm. WT, wild-type mice; 3×Tg, 3×Tg AD mice; BBR + 3×Tg; 3×Tg AD mice were given BBR at 100 mg/kg/day. One-way ANOVA and Dunnett’s post-hoc test were used to analyze all the data, which are presented as mean ± SD; *n* = 12 animals/group. ** *p* < 0.01.

**Figure 4 cells-10-01161-f004:**
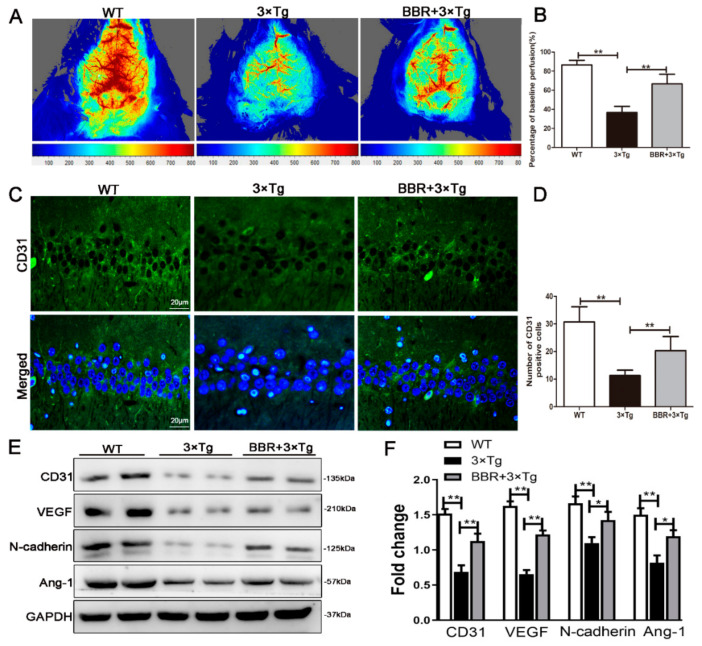
BBR increased the cerebral cortical blood flow perfusion and increased proangiogenic factors in the brains of 3×Tg AD mice. (**A**) Representative images of cerebral cortical blood flow perfusion in three groups. (**B**) Analysis of the level of cerebral cortical blood flow perfusion in three groups. (**C**) Immunofluorescence of CD31 (green) in the hippocampi of mice from the three groups. Scale bar: 20 μm. (**D**) Quantification of CD31 immunoreactivity in the hippocampi of mice. (**E**) Western blotting showed protein level of CD31, VEGF, N-Cadherin and Ang-1 in three groups. (**F**) Analysis of protein level of CD31, VEGF, N-Cadherin and Ang-1. WT, wild-type mice; 3×Tg, 3×Tg AD mice; BBR + 3×Tg; 3×Tg AD mice were given BBR at 100 mg/kg/day. One-way ANOVA and Dunnett’s post-hoc test were used to analyze all the data, which are presented as mean ± SD; *n* = 12 animals/group. * *p* < 0.05, ** *p* < 0.01.

**Table 1 cells-10-01161-t001:** Antibody information.

Antibody	Host	Application	Source	Dilutions
Aβ	Rabbit	WB/IF	Abcam	1:500
GFAP	Rabbit	WB/IF	Abcam	1:500
NeuN	Rabbit	WB/IF	Abcam	1:500
CD31	Rabbit	WB/IF	Abcam	1:500
VEGF	Rabbit	WB	Abcam	1:500
GAPDH	Rabbit	WB	Abcam	1:2500
N-cadherin	Rabbit	WB	Cell signaling	1:1000
Ang-1	Rabbit	WB	Cell signaling	1:1000
caspase3	Rabbit	WB/IF	Cell signaling	1:1000

WB: Western blot blatting; IF: immunofluorescence.

## Data Availability

All data generated or analysed during this study are included in this published article.

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
