# Peer review of "Berberine Improves Cognitive Impairment by Simultaneously Impacting Cerebral Blood Flow and β-Amyloid Accumulation in an APP/tau/PS1 Mouse Model of Alzheimer’s Disease"

_cells, 2021, doi:10.3390/cells10051161_

Round 1

Reviewer 1 Report

The authors show how berberine may have a potential therapeutic effect on Alzheimer's disease. To reach this conclusion, they have evaluated the effect of treatment with this compound to 3xTg-AD mice and have developed a number of experimental techniques. This is a comprehensive article of potential interest. However, there are some small issues that need to be corrected for the article to be published:

The article requires a moderate revision of the English language.

Authors should include references that demonstrate the statements they make in both the introduction and the discussion (An example is in position). The introduction should be more complete.

I recommend that the authors introduce the 3xTg-AD model in the introduction, as its use is one of the highlights of the paper. Are there previous similar studies with this model?

The methodology is very complete, however the authors should provide more information regarding the antibodies used. Again, this is a general recommendation that should be taken into account. The authors can include a table of antibodies if they see it necessary.

Why have the authors used only males? The authors should answer and justify this question, since in the 3xTg-AD model there is some dimorphism in the response to compounds such as berberine (see doi: 10.1016/j.mcn.2018.06.005).

The results are very nice. The authors should review the figures as they have some small design mistake such as the size of the letters used in Figure 3. 

Finally the discussion should discuss the results. In this case it is a summary of the results obtained with hardly any discussion, as shown by the low number of articles referenced in this section. Are there previous studies in other models? Why have mice of this age been used? Have previous studies been done in this model with compounds that have similar effects?...

If the authors address these small questions, the article will be suitable for publication in the journal cells, in my view.

Author Response

To Reviewer #1:

The authors show how berberine may have a potential therapeutic effect on Alzheimer's disease. To reach this conclusion, they have evaluated the effect of treatment with this compound to 3xTg-AD mice and have developed a number of experimental techniques. This is a comprehensive article of potential interest. However, there are some small issues that need to be corrected for the article to be published:

Q#1The article requires a moderate revision of the English language.

Response: We appreciate the reviewer’s comments, this is an important point. We have modified it as required.

Q#2: Authors should include references that demonstrate the statements they make in both the introduction and the discussion (An example is in position). The introduction should be more complete.

Response: We appreciate the reviewer’s comments, this is an important point. We have modified it as required.

Q#3: I recommend that the authors introduce the 3xTg-AD model in the introduction, as its use is one of the highlights of the paper. Are there previous similar studies with this model?

Response: We appreciate the reviewer’s comments, this is an important point. We have introduced the 3xTg-AD model in the Animals and Treatment. This study utilized 3×Tg AD mice expressing APPswe, PS1M146V, and tauP301L human gene mutants, which were purchased from the Jackson Laboratory (Bar Harbor, ME, United States). The intracellular Aβ was detected at the age of 6 months in this mice model. A lot of Alzheimer's studies have used this animal model in the past.

Q#4: The methodology is very complete, however the authors should provide more information regarding the antibodies used. Again, this is a general recommendation that should be taken into account. The authors can include a table of antibodies if they see it necessary.

Response: We appreciate the reviewer’s comments, this is an important point. We have modified it as required.

Q#5: Why have the authors used only males? The authors should answer and justify this question, since in the 3xTg-AD model there is some dimorphism in the response to compounds such as berberine (see doi: 10.1016/j.mcn.2018.06.005).

Response: We appreciate the reviewer’s comments, this is an important point. In this study, we report that chronic administration of Berberine reduces Aβ deposits and cognitive impairments in a well characterized strain of PS1M146V, APPSwe, and tauP301L transgenic mice(3×Tg-AD mice). Berberine were chosen according to previous studies (Weijia Kong, Jing Wei et al, Berberine is a novel cholesterol-lowering drug working through a unique mechanism distinct from statins. Nature medicine 2004,1344-1351) and has no gender differences.

Q#6: The results are very nice. The authors should review the figures as they have some small design mistake such as the size of the letters used in Figure 3. 

Response: We appreciate the reviewer’s comments, this is an important point. We have modified it as required.

Q#7: Finally the discussion should discuss the results. In this case it is a summary of the results obtained with hardly any discussion, as shown by the low number of articles referenced in this section. Are there previous studies in other models? Why have mice of this age been used? Have previous studies been done in this model with compounds that have similar effects?...

Response: We appreciate the reviewer’s comments, this is an important point. We have modified it as required.

Reviewer 2 Report

This study shows that berberine ameliorated vicious phenomena such as the decrease of CBF, the memory impairment, the Aβdeposition, the neuronal cell death and the decrease of proangiogenic factors occurred in 3xTg AD mice. The authors said that the purpose of this study was the test of berberine’s effects and the understanding of the underlying mechanisms. I think that the experiments were implemented well and used adequate methods. And the results seem to be interesting. However I think that this study just made me know the phenomena induced by berberine. I want the authors to discuss about the mechanisms of berberine’s effects. For example it is well known that berberine has several effects as anti-inflammatory or anti-oxidative substance. They might be the underlying effects of the results of this study.

Minor points;

  1. L40: The full word of the abbreviation “VD” should be described.
  2. L66: Berberine in the whole text should be unified. Some are berberine, some berberine in the top of the sentence, some Berberine in the middle of the sentence, and some BBR in the text.
  3. L90: How was berberine administered? Did mice drink in drinking water ad lib? If mice drank ad lib, how was the dose controlled at 100 mg/kg/day?

Or was berberine given per oral in drinking water?

  1. L270: “figure 3D” may be “figure 3G”.
  2. In Fig.3 : What were merged in Fig.3A and C? What are blue in the figure should be described in the figure legend.
  3. In Fig.3: The Red intensity of 3XTg in Fig.3E (photos) seems brighter than those of WT and BBR+3XTg, which does not coincide with Fig.3F.
  4. L275: The first “berberine” should be “Bernerine”.
  5. L293: The following sentence is obscure, “And it could be inferred that the neovascularization promoted by BBR……………….”
  6. Fig.4C: Please describe what are merged.
  7. L391: “VEFG” should be “VEGF”.
  8. L398: Put “.” after “pericytes”.
  9. L411: The following sentence is obscure, “It is indicated that berberine-promoted neovascularization……………….”
  10. L418: The “ameliorating” should be “ameliorate”.
  11. I think that “Conclusion” here seems speculative and kind of literature. The authors might want to draw more specific conclusion.

Author Response

To Reviewer #2:

Comments and Suggestions for Authors

This study shows that berberine ameliorated vicious phenomena such as the decrease of CBF, the memory impairment, the Aβdeposition, the neuronal cell death and the decrease of proangiogenic factors occurred in 3xTg AD mice. The authors said that the purpose of this study was the test of berberine’s effects and the understanding of the underlying mechanisms. I think that the experiments were implemented well and used adequate methods. And the results seem to be interesting. However I think that this study just made me know the phenomena induced by berberine. I want the authors to discuss about the mechanisms of berberine’s effects. For example it is well known that berberine has several effects as anti-inflammatory or anti-oxidative substance. They might be the underlying effects of the results of this study.

Minor points;

Q#1: L40: The full word of the abbreviation “VD” should be described.

Response: We have corrected it in the revised manuscript. Vascular dementia (VD).

Q#2: L66: Berberine in the whole text should be unified. Some are berberine, some berberine in the top of the sentence, some Berberine in the middle of the sentence, and some BBR in the text.

Response: We have corrected them in the throughout manuscript. Berberine (BBR).

Q#3: L90: How was berberine administered? Did mice drink in drinking water ad lib? If mice drank ad lib, how was the dose controlled at 100 mg/kg/day? Or was berberine given per oral in drinking water?

Response: we appreciate the reviewer’s comments. Every 3 mice of the same gender with approximate weight were put into one cage from newly born till 4 months old. Measure the water consumption of every cage every day, from which can count the daily water consumption of every cage of mice from the previous data of 4 months. According to the weight of the mouse, take 100mg /Kg /day as a standard to calculate the amount of Berberine for one mice needed for per day and put the Berberine into water by ultrasonic oscillations. The does of Berberine were chosen according to previous studies (Weijia Kong, Jing Wei et al, Berberine is a novel cholesterol-lowering drug working through a unique mechanism distinct from statins. Nature medicine 2004,1344-1351) and has no gender differences. The method of putting drug into water to feed animals is recognized by the international community.

Q#4: L270: “figure 3D” may be “figure 3G”.

Response: Thank you for your kindly remind. We have corrected it in the revised manuscript.

Q#5: In Fig.3: What were merged in Fig.3A and C? What are blue in the figure should be described in the figure legend.

Response: Thank you for your kindly remind. We have corrected it in the revised manuscript. The blue in the figure is DAPI.

Q#6: In Fig.3: The Red intensity of 3XTg in Fig.3E (photos) seems brighter than those of WT and BBR+3XTg, which does not coincide with Fig.3F.

Response: Thank you for your kindly remind. We have corrected it in the revised manuscript.

Q#7: L275: The first “berberine” should be “Bernerine”.

Response: Thank you for your kindly remind. We have corrected it in the revised manuscript.

Q#8: L293: The following sentence is obscure, “And it could be inferred that the neovascularization promoted by BBR……………….”

Response: Thank you for your kindly remind. We have corrected it in the revised manuscript. It can be inferred that BBR promotes the formation of blood vessels, and the new vessels have blood flow through, and it is functional blood.

Q#9: Fig.4C: Please describe what are merged.

Response: Thank you for your kindly remind. We have corrected it in the revised manuscript. CD31 and DAPI are merged.

Q#10: L391: “VEFG” should be “VEGF”.

Response: Thank you for your kindly remind. We have corrected it in the revised manuscript.

Q#11: L398: Put “.” after “pericytes”.

Response: Thank you for your kindly remind. We have corrected it in the revised manuscript.

Q#12: L411: The following sentence is obscure, “It is indicated that berberine-promoted neovascularization……………….”

Response: Thank you for your kindly remind. We have corrected it in the revised manuscript. It is indicated that berberine promoted angiogenesis,which improved blood perfusion of brain tissue.

Q#13: L418: The “ameliorating” should be “ameliorate”.

Response: Thank you for your kindly remind. We have corrected it in the revised manuscript.

Q#14: I think that “Conclusion” here seems speculative and kind of literature. The authors might want to draw more specific conclusion.

Response: Thank you for your kindly remind. We have corrected it in the revised manuscript. The BBR can improve the learning and memory abilities of the AD model mice obviously. The BBR’s protection on AD may be contected with promoting CBF recovery and repress the pathology of AD. Taking all these results into consideration, it could be concluded that BBR could promote angiogenesis, accelerate the elimination of toxic substances such as AB in the tissue structure around microvessels, and inhibit the apoptosis of nerve cells. These results suggest that BBR could have a potential for the therapy of Alzheimer’s disease through the promoting CBF recovery.

Reviewer 3 Report

Authors should respond to this points before the manuscript could be considered for publication.

Abbreviation list and explanation must be added.

lane 90: 7g/ di in 70 kg is a very high dose. Why?  Implications in Human?

lane 155: "paraffin sections of mouse brain were...5mm thickness" ? The glass or the tissue sections?

lane 255: What GFAP data mean?

Figure 1: They have to compare statistically WT and BBR+3xTG.

Figure 2: Graphic symbols are difficult to differentiate.

Figure 3: Barberine mitigates but not entirely suppresses AD Neurotoxins and Apoptosis. I think they have to stress this aspect in their manuscript. 

Discussion:

lane 329: p-TAU is ignored, they have to add it.

Describe first your scientific data and then the cognitive tests.

lane 366-367: changes in GFAP expression are not discussed.

lane 391: VEGF behavior in AD mice likely differs from that in human AD, please discuss it.

Author Response

To Reviewer #3:

Comments and Suggestions for Authors

Authors should respond to this points before the manuscript could be considered for publication.

Q#1: Abbreviation list and explanation must be added.

Response: Thank you for your kindly remind. We have corrected it in the revised manuscript.

Q#2:lane 90: 7g/ di in 70 kg is a very high dose. Why?  Implications in Human? 7g/ di

Response: we appreciate the reviewer’s comments. The equivalent dose ratio between humans and animals is based on body surface area. If the human clinical dose is X mg /Kg, the mouse dose is 9.1*X mg /Kg. The dose of Berberine were chosen according to previous studies (Weijia Kong, Jing Wei et al, Berberine is a novel cholesterol-lowering drug working through a unique mechanism distinct from statins. Nature medicine 2004,1344-1351) and has no gender differences.

Q#3: lane 155: "paraffin sections of mouse brain were...5mm thickness" ? The glass or the tissue sections?

Response: Thank you for your kindly remind. The tissue sections were 5mm in thickness.

Q#4: lane 255: What GFAP data mean?

Response: we appreciate the reviewer’s comments. In the pathological state, Astrocyte AS is activated by Aβ and other toxic substances, which can produce cytoinitis factors, complements, oxygen free radicals, etc. and initiate inflammatory reactions, promote damage and death of nerve cells, and aggravate the process of AD. Astrocyte pathological changes can cause damage to vascular regulation and small arterial contraction and hypoxia of tissues, which plays an important role in the pathogenesis of AD. The experimental study showed that compared with the WT group, the expression of GFAP in hippocampus of 3×Tg AD group increased significantly, indicating that the damage to hippocup tissue in the 3×Tg AD group was obvious, and the ascension was active compared with the 3×Tg AD group, and the expression of hippocome GFAP in the berberine group was significantly reduced, indicating that the berberine group inhibited the excessive growth of AD mice AS. This may be related to the improvement of local blood supply by berberine, reducing the toxicity of Aβ, and thus repairing damage to nerve vascular units.

Q#5: Figure 1: They have to compare statistically WT and BBR+3xTG.

Response: Thank you for your kindly remind. We have corrected it in the revised manuscript.

Q#6: Figure 2: Graphic symbols are difficult to differentiate.

Response: Thank you for your kindly remind. We have corrected it in the revised manuscript.

Q#7: Figure 3: Barberine mitigates but not entirely suppresses AD Neurotoxins and Apoptosis. I think they have to stress this aspect in their manuscript.

Response: Thank you for your kindly remind. We have corrected it in the revised manuscript.

Discussion:

Q#8: lane 329: p-TAU is ignored, they have to add it.

Response: Thank you for your kindly remind. We studied p-TAU in another article, so we didn't discuss it in this article.

Q#9: Describe first your scientific data and then the cognitive tests.

Response: Thank you for your kindly remind. We have corrected it in the revised manuscript.

Q#10: lane 366-367: changes in GFAP expression are not discussed.

Response: Thank you for your kindly remind. We have corrected it in the revised manuscript. In the pathological state, Astrocyte AS is activated by Aβ and other toxic substances, which can produce cytoinitis factors, complements, oxygen free radicals, etc. and initiate inflammatory reactions, promote damage and death of nerve cells, and aggravate the process of AD. Astrocyte pathological changes can cause damage to vascular regulation and small arterial contraction and hypoxia of tissues, which plays an important role in the pathogenesis of AD. The experimental study showed that compared with the WT group, the expression of GFAP in hippocampus of 3×Tg AD group increased significantly, indicating that the damage to hippocup tissue in the 3×Tg AD group was obvious, and the ascension was active compared with the 3×Tg AD group, and the expression of hippocome GFAP in the berberine group was significantly reduced, indicating that the berberine group inhibited the excessive growth of AD mice AS. This may be related to the improvement of local blood supply by berberine, reducing the toxicity of Aβ, and thus repairing damage to nerve vascular units.

Q#11: lane 391: VEGF behavior in AD mice likely differs from that in human AD, please discuss it.

Response: Thank you for your kindly remind. We have corrected it in the revised manuscript. VEGF can act directly on many types of nerve cells to play a neurotrophic and neuroprotective role, enhance cell activity and survival, and promote axon regeneration. The experimental study shows that compared with the WT group, the expression of VEGF in hippocampus of the 3×Tg AD group mice was significantly decreased, indicating that 3×Tg AD group mice have vascular damage and neovascular regeneration disorders, and thus lead to nerve vascular unit damage and brain vascular reserves, microcirculation blood supply is poor, and thus cognitive dysfunction, which may be related to the nerve and vascular toxicity of Aβ. And Compared with 3×Tg AD group, VEGF in Berberine group was significantly increased, which was related to berberine promoting angiogenesis and the acceleration of the removal of toxic substances such as Aβ by the tissue structure around microvascular vessels. Moreover, angiogenesis is necessary for local tissue metabolism and functional recovery, which can increase cerebral blood flow and improve learning and memory ability.
